# CircRNAs in Tumor Radioresistance

**DOI:** 10.3390/biom12111586

**Published:** 2022-10-28

**Authors:** Yining Gao, Jiawen Gao, Fei Lin, Ting Wang, Sitong Huo, Jiefang Wu, Qi Zhou, Chao Zhang

**Affiliations:** 1Department of Biochemistry and Molecular Biology, School of Basic Medical Science, Southern Medical University and Guangdong Provincial Key Laboratory of Single Cell Technology and Application, Guangzhou 510000, China; 2Department of Biochemistry and Molecular Biology, School of Basic Medical Science, Southern Medical University and School of Public Health, Southern Medical University, Guangzhou 510000, China; 3Division of Spine Surgery, Department of Orthopaedics, Nanfang Hospital, Southern Medical University, Guangzhou 510000, China; 4School of Medicine, Tsinghua University, Beijing 100084, China; 5Department of Radiology, Nanfang Hospital, Southern Medical University, Guangzhou 510515, China; 6Department of Cardiology, the Fifth Affiliated Hospital of Southern Medical University, 1838 Guangzhou Boulevard, Guangzhou 510000, China

**Keywords:** circular RNAs, miRNAs, tumor, radioresistance, radiation therapy

## Abstract

**Circular** RNAs (circRNAs) are endogenous, non-coding RNAs, which are derived from host genes that are present in several species and can be involved in the progression of various diseases. circRNAs’ leading role is to act as RNA sponges. In recent years, the other roles of circRNAs have been discovered, such as regulating transcription and translation, regulating host genes, and even being translated into proteins. As some tumor cells are no longer radiosensitive, tumor radioresistance has since become a challenge in treating tumors. In recent years, circRNAs are differentially expressed in tumor cells and can be used as biological markers of tumors. In addition, circRNAs can regulate the radiosensitivity of tumors. Here, we list the mechanisms of circRNAs in glioma, nasopharyngeal carcinoma, and non-small cell lung cancer; further, these studies also provide new ideas for the purposes of eliminating radioresistance in tumors.

## 1. Introduction

CircRNAs were first discovered, in 1976, in RNA viruses [1]. With the development of high-throughput RNA sequencing and bioinformatics tools, scientists have discovered that circRNAs are commonly found in humans and many other animals [2,3], including fungi, protozoa, plants, worms, fish, insects, and mammals [4]. CircRNAs are derived from host genes and are found primarily in the cytoplasm [5]. Similar to linear mRNAs, circRNAs are derived from linear precursor mRNAs (i.e., pre-mRNAs), which are transcribed by RNA polymerase II [6]—relying on canonical splicing machinery—including splice signal sites and spliceosomes [7]. Due to the lack of 5’ end cap or 3’ poly(A) tail, circRNAs are covalently closed endogenous biomolecules that fall into non-coding RNA (ncRNA) molecules [8]. The unique structure of circRNAs allow for a longer half-life and a more excellent resistance to RNase R than linear RNAs [9], making it a potential diagnostic biomarker and therapeutic target [10].

For a long time, circRNAs were considered to be “non-coding” RNA with regulatory effects [11]. After the discovery of translatable circRNAs, attention again focused on this particular structure [12]. Increasingly, studies have identified additional functions for circRNAs, including acting as miRNA sponges [13] or protein scaffolds [2], as well as being translated into peptides [14]. Scientists have found that these RNAs have tissue-specific, cell-specific, and developmental stage-specific expression patterns [15]; further, circRNAs are also conserved across species.

Radiotherapy is one of the main methods of oncology treatment and is defined as the application of radiation to kill tumor cells or control their proliferation in clinical cancer treatment [16]. Through direct and indirect mechanisms, X-rays penetrate the tumor tissue and induce cytotoxic damage to proliferating cells [17]. The radiobiological phenomena are summarized as the “4Rs of radiotherapy”, i.e., repair, redistribution/recombination, repopulation, and reoxygenation, which are together the basis of fractionated radiotherapy [18]. These four phenomena are often extended by a fifth ‘R’, namely, intrinsic radioresistance, which is defined as radiation-induced initial DNA damage [19].

## 2. Biogenesis and Functional Mechanisms of CircRNAs

### 2.1. Biogenesis of CircRNAs

CircRNAs can be the main product generated from the host gene [15], for instance, the human cytochrome P450 gene, the rat androgen-binding protein (ABP) gene [20], the human dystrophin gene [21], and the human inhibitor of cyclin-dependent kinase 4 (INK4/ARF)-associated non-coding RNA [22]. CircRNAs can be classified as exonic circRNAs (ecircRNAs), intronic circRNAs (ciRNAs), or exon–intron circular RNAs (EIciRNAs). Most circRNAs are ecircRNAs, accounting for over 80% of known circRNAs [23], and they are mainly located in the cytoplasm [24,25], while EIcircRNAs and ciRNAs are usually found in the nucleus [26]. Exon-derived circRNAs are produced by a specific type of splicing known as post-snap [6]. Splice sites with specifically canonical splice sites accomplish post splicing [20,27]. In this type of splicing, the 5’ splice donor attacks the upstream 3’ splice site. This results in a 3’−5’phosphodiester bond that generates a circular RNA molecule [28]. There are also circRNAs that are formed by intron pairing, which are formed by the bringing of the splice donor site and the upstream splice acceptor site into proximity in order to form a loop via reverse complementary sequences, such as ALU repeats [29]. In the RNA-binding protein (RBP)-mediated model, the specific trans-activator RBP binds specifically to each flanking intron, forming a bridge that brings the splice donor and acceptor sites close enough to form a loop [30] (Figure 1).

### 2.2. Functions of CircRNAs

#### 2.2.1. MiRNA Sponges

Acting as miRNA sponges is the most reported function of circRNAs [31]. The initial observation that some circRNAs have many miRNA binding sites led to speculation that these molecules may act as miRNA sponges [32]. MiRNAs are small non-coding RNAs that bind to target mRNAs and typically induce mRNA degradation or translational repression [33,34]. Many circRNAs have been found to bind miRNAs extensively, reducing their effectiveness, and thus upregulating the expression of their target mRNAs [35,36].

#### 2.2.2. CircRNAs Regulate Transcription and Translation

CircRNAs can directly bind to their parental mRNAs, thereby affecting protein translation [6]. Competition for the RNA-binding protein HuR by circRNAs and their cognate mRNAs has also been reported to affect protein expression [37]. In addition, circRNAs can influence ribosome function and affect protein synthesis [38]. For example, inter-exon retained intronic circRNAs (exon–intron circRNAs and EIciRNAs) can bind to U1 small nuclear ribonucleoproteins via RNA–RNA interactions between snRNAs and EIciRNAs, and then interact with Pol II at the parental gene promoter in order to enhance their expression [26]. Similarly, cyclic intron RNA (ciRNAs) formed by detached unbranched lassoes can accumulate at their synthesis sites and increase the expression of the parental gene by regulating the prolonged Pol II activity [39].

#### 2.2.3. CircRNAs Interact with Proteins

CircRNA–protein interactions are another vital function of circRNAs. RBP is a class of proteins associated with RNA metabolism. These proteins are involved in forming ribonucleoprotein complexes by mediating the maturation, translocation, localization, and translation of RNA [40]. It has been reported that RNA–protein interactions influence protein expression and function and regulate the synthesis and degradation of circRNAs [41]. Some CircRNAs have binding sites for proteins and can effectively act as protein sponges [42,43]. CircRNAs can also act as protein decoys, cooperating with target proteins at appropriate locations in the cell in order to alter the conventional physiological functions of the protein. In addition, circRNAs can act as scaffolds to facilitate contact between two or more proteins, promote co-localization of enzymes and their substrates, or facilitate nuclear translocation, thereby influencing the cell cycle [44]. It has also been suggested that circRNAs may recruit specific proteins to certain locations in the cell, although the exact mechanisms are still not understood [45].

#### 2.2.4. Translation of CircRNAs into Proteins

CircRNAs were initially thought to be untranslatable. However, later studies have shown that circRNAs can be translated both in vitro and in vivo [46]. Studies of circRNA encoding proteins have shown that internal nuclear protein entry sites (IRES) and open reading frames (ORFs) are essential components of circRNA protein translation [47,48]. Due to the lack of 5’-cap and 3’-tail, circRNAs can only adopt a cap-independent approach [49]. The microproteins encoded by circRNAs are relatively short, ranging from 146–344 amino acids in length. Almost all circRNA-encoded proteins are found in metabolically active cells such as cancer cells or myogenic cells [50]. The translated proteins may also have some function in these cells, although the physiological processes of most of these proteins have not yet been determined [51].

#### 2.2.5. Exosomal CircRNAs

Exosomes are 40–200 nm diameter structures with lipid bilayer membranes, and almost all cell types can secrete them [52,53]. Exosomes can contain various substances such as proteins, lipids, DNA, and RNA [54]. When released and transferred to recipient cells, they can participate in intercellular communication [55,56]. The presence of large and stable circRNAs in exosomes, and their assistance in the clearance of circRNAs, provides evidence for the degradation of circRNAs [57,58]. Researchers have recently found that extra circRNAs may be transported to immune cells as tumor antigens, activating anti-tumor immunity, or binding to miRNAs and proteins, thereby modulating immune cell activity. In addition, when exocircRNAs are transported from tumor cells to immune cells, they contribute to the release of miRNAs into immune cells, silencing relevant target genes as a result [59].

## 3. Research Progress of CircRNAs Related to Tumor Radioresistance

When tumor cells are genetically or phenotypically affected by radiation exposure, or protected from treatment by the tumor microenvironment, cancer cells begin to acquire resistance to radiation, resulting in a diminished effect of radiotherapy [60]. Many previous studies have been focused on enhancing sensitivity to radiotherapy, such as targeting DNA damage and repair, modulating growth factors, affecting tumor stem cells, and generating reactive oxygen species [61]. Here, we list dozens of reports on the role of circRNAs in tumor radioresistance (Table 1). According to these studies, circRNAs can regulate autophagy, epithelial-to-mesenchymal transition (EMT), and function as exosomal circRNAs (Figure 2). By understanding the impact of circRNAs on tumor radioresistance, new therapeutic insights into tumors may be gained.

### 3.1. Glioma

Glioma originates from glial cells (precursors) and is one of the most common malignant tumors of the central nervous system (CNS) [89]. It has an incidence of about 5–10/100,000 in the population and has been increasing yearly in recent years [90,91]. The 5-year survival rate for the most common of gliomas, glioblastoma, is only about 5% [92]. Treatment of gliomas is primarily surgical resection, with radiation therapy routinely administered after surgery due to the tumors’ aggressive growth [93,94]. Most gliomas are insensitive to radiotherapy [95], and increasing the dose of X-rays can cause irreversible damage to normal glial cells [96]. Therefore, we need to gain insight into the molecular radiobiological mechanisms of glioma resistance to radiation, which is essential to improve the sensitivity of glioma radiotherapy and to reduce the damage to normal brain tissue.

Zhao et al. [62] found that circ-0008344 was highly expressed in radioresistant tissues and therefore hypothesized that circ-0008344 was associated with radioresistance in glioma. Downregulation of the circ-0008344 gene enhanced apoptosis, DNA damage, and the enhanced radiosensitivity of glioma. miR-433-3p could stimulate radiosensitivity of glioma cells; further, via target gene analysis of circ-0008344 and miR-433-3p, the results showed that circ-0008344 contains a binding site for miR-433-3p and exerts a sponge effect on miR-433-3p in glioma cells. In addition, RNF2 was identified as a downstream gene of miR-433-3p. Thus circ-0008344 was shown to increase RNF2 expression by acting as a miR-433-3p sponge. RNF2 overexpression rescued circ-0008344 downregulation of radiosensitivity to glioma cells. In a study by Zhu et al., downregulation of circ-VCAN inhibited proliferation, the migration and invasion of glioma cells, and enhanced apoptosis. Furthermore, circ-VCAN was negatively correlated with miR-1183 expression and circ-VCAN negatively regulated miR-1183 via direct binding. This implies that circ-VCAN accelerated proliferation, migration and invasion, and also inhibited the apoptosis of irradiated glioma cells via regulating miR-1183 [63].

CircRNAs can also influence tumor radioresistance by affecting glycolysis [64]. Glycolysis is a process that converts glucose to pyruvate and then to lactate, which provides cellular energy and participates in macromolecular biosynthesis [97]. Tumor cells, and even various normal cells, tend to exhibit a high rate of glycolysis, also called the “Wartburg effect,” regardless of oxygen availability [98]. The glycolytic process is usually accompanied by glucose uptake, lactate production, and ATP production, which is an essential determinant of cellular drug resistance [97]. Guan et al. showed that circ-PITX1 expression was upregulated in glioma tissues and cells compared to normal tissues. Further, the lack of circ-PITX1 inhibited glioma cell viability, glycolysis, the ability to form radiation-resistant clones in vitro, and tumor growth in vivo—suggesting that circ-PITX1 is a positive regulator of glioma development. The expression of miR-329-3p was decreased in glioma samples compared to the control, and miR-329-3p expression was negatively correlated with circ-PITX1 in glioma samples, suggesting that the oncogenic effect of circ-PITX1 was achieved by sponging miR-329-3p. In the study of Guan et al., NEK2 was shown to be a direct target of miR-329-3p. A previous study also found that NEK2 was expressed and enriched in glioma. Further, miR-128 targeted to regulate apoptosis in glioma cells [99]. By introducing both the overexpression vector circ-PITX1 and the glycolysis inhibitor 2-DG into glioma cells, it was demonstrated that circ-PITX1 overexpression promoted the glycolytic process and radiation resistance, but 2-DG counteracted this promotion. It was shown that circ-PITX1 knockdown inhibited glycolysis and made glioma cells sensitive to radiation treatment.

In addition to affecting the radioresistance of gliomas, circRNAs can also influence the proliferation and invasion of gliomas through tumor-associated signaling pathways. Aberrantly expressed, circRNAs play an important role in glioma proliferation, cell cycle, invasion, and metastasis via regulating complementary miRNAs or targeting mRNA that are related to tumor-associated signaling pathways, such as PI3K/AKT/mTOR and the Wnt/β-catenin pathway [100]. For example, the silencing of circ-ZNF292 inhibits glioma proliferation and cell cycle progression via the Wnt/β-catenin pathway [101]. Circ-0014359 and circ-NT5E promote glioma progression via the PI3K/AKT/mTOR signaling pathway via miR-153 and miR-422a, respectively [102,103]. Therefore, understanding the cancer-related pathways regulated by circRNAs will also provide new therapeutic ideas for radiotherapy resistance in glioma.

### 3.2. Nasopharyngeal Carcinoma

Nasopharyngeal carcinoma (NPC) is an epithelial malignancy whose occurrence may be linked to viral, environmental, and genetic factors [104,105]. Nasopharyngeal cancer is endemic in Asian and North African populations. Most patients in China are found in southern regions, such as Guangdong and Guangxi [106]. Nasopharyngeal cancer has a high degree of malignancy, a high incidence of metastasis, early metastasis, and no obvious specific symptoms in the early stages. Most patients with nasopharyngeal carcinoma do not seek treatment until the middle or late stages; further, this type of cancer has a low 5-year survival rate [107]. Although radiotherapy can improve the 5-year survival rate of patients with nasopharyngeal cancer, some patients may experience local recurrence and distant metastases after receiving radiotherapy due to the limitations of radioresistance [108].

In a study by Chen et al. [65], circRNA-000543 was highly expressed in radiation-resistant nasopharyngeal carcinoma tissues and radiation-resistant cell lines. Further studies found that circRNA-000543 downregulation promoted the sensitivity of nasopharyngeal carcinoma cells to radiotherapy. In addition, miR-9 expression in nasopharyngeal carcinoma tissues was negatively correlated with circRNA-000543 levels. They also confirmed that PDGFRB is a target of miR-9 in NPC. miR-9 is an important therapeutic target for cancer due to its involvement in angiogenesis, epithelial-mesenchymal transition (EMT), and metastasis. In this study, PDFGRB knockdown reversed miR-9 inhibitor-mediated irradiation resistance. Imatinib, a PDFGRB inhibitor used for CML treatment, sensitized radiation-resistant nasopharyngeal carcinoma cells to irradiation. Pearson analysis showed that miR-9 levels were negatively correlated with circRNA-000543 and PDFGRB expression, while PDFGRB levels were positively correlated with circRNA-000543 expression. Therefore, circRNA-000543 knockdown could sensitize nasopharyngeal carcinoma cells to radiation by targeting the miR-9/ PDGFRB axis.

In addition to circRNA-000543, circ-CCNB1 has also been shown to be associated with EMT [109]. Circ-CCNB1 inhibits migration and invasion of nasopharyngeal carcinoma by promoting binding between NF90 and TJP1 mRNA, stabilizing TJP1 mRNA, and enhancing tight junctions between tumor cells. TJP1 is a key regulator of tight junction assembly and inhibits migration and invasion by coordinating the assembly or dynamics of the cortical cytoskeleton to regulate the function [110]. By revealing the mechanism by which circ-CCNB1 regulates the migration and invasion of nasopharyngeal carcinoma, it may provide a potential marker and target for the diagnosis and treatment of patients with nasopharyngeal carcinoma.

### 3.3. Non-Small Cell Lung Cancer

Non-small cell lung cancer (NSCLC) includes squamous cell carcinoma, large cell carcinoma, and adenocarcinoma. It accounts for about 80% or more of all primary lung cancer cases [111,112,113]. Most patients are diagnosed with advanced tumors and have a 5-year survival rate of less than 20% [114]. Radiotherapy is currently the primary method of non-surgical treatment for NSCLC. Radiotherapy damages the DNA of tumor cells and causes cell death [115]. However, the effectiveness of radiotherapy is limited during therapy due to the tumors’ acquired radioresistance, resulting in a poor prognosis [116,117].

Zhang et al. [68] demonstrated by luciferase reporter gene assay, RIP assay, and qRT-PCR that circ-0001287 sponges miR-21 and negatively regulates its expression. MiR-21 is considered an oncogene in many cancers, enhances cell proliferation, metastasis, and radiation resistance, and inhibits apoptosis in NSCLC cells [118,119]. MiR-21 is one of the most critical regulators of PTEN expression [120,121]. The PI3K signaling pathway is one of the most critical pathways in tumor biology, regulating cell cycle progression, survival, migration, invasion, and tumor cell metabolism. PTEN is its primary negative regulator, which it achieves via dephosphorylating PIP3 to PIP2. PTEN also acts as a protein phosphatase regulating chromosome stability, DNA repair, and apoptosis [122,123]. Circ-0001287 sponge miR-21, in turn, upregulates PTEN expression and inhibits NSCLC cell proliferation, metastasis, and radiation resistance.

Kim et al. [69] also validated the mode of action of one of the circRNA–miRNA–mRNA networks in the progression of NSCLC. They explored the role of circ-0086720 in NSCLC cells after radiotherapy and found that downregulation of circ-0086720 gene enhanced radiosensitivity, further impaired cell survival, and induced apoptosis. It was also demonstrated that miR-375, a target of circ-008672, was reduced in expression in radiation-resistant NSCLC tissues. Further analysis revealed that the word of SPIN1 in NSCLC cells decreased after circ-0086720 knockdown, while the reintroduction of miR-375 inhibitor increased the expression of SPIN1.

Immune escape in NSCLC cells has also been associated with circRNAs. PD1 is a negative costimulatory receptor that plays a crucial role in suppressing T cell activation in vitro and in vivo [124]. For example, circIGF2BP3 alleviates the inhibitory effect of miR-328-3p and miR-3173-5p on PKP3 expression, which is achieved by acting as a miRNA sponge. Additionally, PKP3 stabilizes PD-L1 in an otub1-dependent manner. The circIGF2BP3/PKP3 axis is ultimately involved in the immune escape of NSCLC cells by upregulating PD-L1 expression [125]. Increased expression of circ-USP7 reduced the efficacy of anti-PD1 therapy through the exosomal circUSP7/miR-934/SHP2 axis [126]. These studies suggest that circRNAs may not only be used as tumor markers, but may also influence the critical factors that are present in treating patients with NSCLC.

### 3.4. Colon Cancer and Colorectal Cancer

Both colon and colorectal cancers are the most common digestive system malignancies and pose a heavy burden on human health worldwide [127]. Unhealthy dietary patterns and genetic factors are considered to be risk factors for the development of colon cancer and colorectal cancer [128]. Further, because early symptoms are not obvious, most patients are already at an advanced stage at diagnosis [129]. Radiotherapy is one of the main options for treating colorectal cancer—alone or in combination with surgery and chemotherapy [130,131]. Unfortunately, the resistance of cancer cells to radiation still limits the effectiveness of radiotherapy. Several factors—for example, dysregulated radiosensitivity-related gene expression—are known to affect cellar radiosensitivity [132]. There is growing evidence that circRNAs play an increasingly important role in regulating radiation response [133].

Gao et al. [72] showed that circ-0055625 expression was significantly upregulated in colon cancer tissues and that down-regulation of the circ-0055625 gene inhibited cell proliferation, migration, and invasive ability. MSI1 has been shown to have a role in the development of colon cancer [134]. MSI1 expression was significantly upregulated in colon cancer tissues and cells and increased in colon cancer cells following radiotherapy. In addition, MSI1 overexpression attenuated the effect of circ-0055625’s deletion on colon cancer tumor development and radiosensitivity. This evidence suggests that downregulation of the circ-0055625 gene inhibits colon carcinogenesis’ development and increases the sensitivity of colon cancer to IR by regulating MSI1. Further, miR-338-3p, a tumor suppressor, promotes radiosensitivity in colon cancer. The data showed circ-0055625 was associated with miR-338-3p and miR-338-3p binding to MSI1. Furthermore, the results explain that deletion of circ_0055625 down-regulates MSI1 expression by binding to miR-338-3p. In another study on colon cancer [73], knockdown of circ-CCDC66 also improved the radiosensitivity of colon cells by upregulating miR-338-3p expression, providing a potential therapeutic target for patients with radioresistant colon cancer.

The transmission of circRNA is usually achieved via exosomes [135]. Exosomal information of circRNA is involved in the development of carcinogenesis and radioresistance in humans [75,136]. Circ-0067835 can be transmitted via exosomes, and upregulation of circ-0067835 is associated with CRC development and radioresistance. Wang et al. demonstrated that circ-0067835 directly targets miR-296-45p and that by upregulating miR-296-5p, knockdown of circ-0067835 inhibited CRC development and enhanced cellular radiosensitivity in vitro. The data first show that miR-296-5p directly targets IFG1R, a transmembrane receptor belonging to the tyrosine kinase family [137,138]. In addition to circ-0067835, circ_MFN2 is also a circRNA upregulated in CRC [76]. Circ-MFN2 can promote proliferation, metastasis, and radiation resistance in CRC by regulating the miR-574-3p/IGF1R axis, suggesting that circ-MFN2 may be an oncogene in CRC.

Some cancer-related signaling pathways can be regulated by circRNA-encoded proteins, which is unexpected. Further, circPPP1R12A is expressed at an increased rate in colon cancer. But it is not circPPP1R12A that actually promotes the growth and metastasis of colon cancer cells, but circPPP1R12A-73aa, a short peptide chain encoded by circPPP1R12A. PPPP1R12A is primarily involved in the RhoA/ROCK signaling pathway, which regulates cell adhesion, motility, proliferation, differentiation, and apoptosis [139].

### 3.5. Esophageal Cancer

Esophageal cancer is the eighth most common cancer in the world and the sixth most common cause of cancer death [140]. Current treatment for esophageal cancer includes surgery, radiotherapy, as well as chemotherapy and their combinations [141]. However, due to the aggressive nature of esophageal cancer and the lack of early diagnostic markers, patients’ prognosis is poor, with a 5-year survival rate of only about 20% [142]. Radiotherapy plays a crucial role in treating esophageal cancer, and resistance to radiotherapy is thought to be a significant cause of treatment failure and local tumor recurrence [143]. Therefore, it is essential to explore the molecular mechanisms of radiotherapy resistance in order to improve the prognosis of patients with esophageal cancer.

High expression of circRNA-100367 was associated with the radiosensitivity of ESCC. Silencing circRNA-100367 reduced proliferation and migration of KYSE-150R cells in vitro and inhibited tumor growth in vivo. In addition, miR-217/Wnt3 was shown to be a downstream target of circRNA-100367 regulating radiosensitivity of ESCC. Wnt3, a member of the Wnt family, has been shown to promote the stabilization of β-catenin and regulate the radiosensitivity of cancer cells [144]. By silencing Wnt3, Liu et al. altered the phenotype of KYSE-150R cells and inhibited colony formation and migration of KYSE-150R cells, suggesting that Wnt3 could inhibit the radiosensitivity of ESCC cells.

In the study by He et al. [82], overexpression of circ-VRK1 effectively attenuated ESCC proliferation, migration, and EMT processes. Based on RIP and luciferase reporter gene assays, miR-624-3p interacted with circVRK1. Further analysis revealed that circ-VRK1 enhanced the expression of PTEN, a suppressor of the PI3K/AKT signaling pathway, through the sponge uptake of miR-624-3p [145,146]. Western blotting showed that circ-VRK1 overexpression increased PTEN levels and decreased p-PI3K, p-AKT, and p-mTOR levels. The results suggest that circ-VRK1 inhibits the progression and radiation resistance of ESCC by upregulating PTEN to reduce PI3K/AKT signaling pathway activity.

CircRNA can also regulate the progression of esophageal squamous cell carcinoma in many other ways. Miyagi et al. (101) demonstrated that circ-PUM1, a nuclear genomic-derived circRNA localized to mitochondria, responds to induction of HIF1α protein levels and increases with the accumulation of HIF1α in CoCl2-treated cells. Further, circ-PUM1 may act as a scaffolding protein for mitochondrial complex III assembly and regulates mitochondrial oxidation through the interaction with UQCRC2 phosphorylation. In addition, circ-PUM1 may promote tumor growth by inhibiting focal degeneration in ESCC cells. This suggests that knocking down certain circRNAs may reduce radioresistance in nasopharyngeal carcinoma.

### 3.6. Prostate Cancer

Prostate cancer (PCa) is second only to lung cancer in men, with a high prevalence of 13.5% [127]. Radiation therapy has made exciting breakthroughs in prostate cancer metastasis and post-operative repair [147]. An in-depth understanding of the mechanisms of radioresistance in prostate cancer may help to treat recurrent prostate cancer and interrupt its metastasis [148].

Circ-CCNB2 is overexpressed in irradiation-resistant PCa tissues and cells [85]. Knockdown of circ-CCNB2 suppressed colony formation and metastasis, and also promoted apoptosis in irradiation-resistant PCa cells. Autophagy is a cellular process that regulates cell signaling and maintains endocytosis [149], and the regulation of autophagy is closely related to the radiosensitivity of cancer cells [150]. Cai et al. found a facilitatory effect of autophagy after detecting the associated proteins in radiation-resistant PCa tissues and cells, suggesting that autophagy may induce the formation of radiation resistance in PCa. They then found that the autophagy inhibitor 3-MA restored the pro-tumor effects of circ-CCNB2 overexpression on irradiation-resistant PCa cells, demonstrating the role of circ-CCNB2 in regulating the radiosensitivity of PCa, and which was attributed to autophagy. Furthermore, they determined that circ-CCNB2 regulates radiosensitivity by targeting miR-30b-5p. KIF18A, a downstream target of miR-30b-5p, is an oncogene that promotes the growth and metastasis of lung adenocarcinoma cells [151,152]. Thus, KIF18A can abrogate the inhibitory effects of miR-30b-5p on cell growth, metastasis and autophagy, suggesting that KIF18A also plays a pro-tumor role in irradiation-tolerant PCa cells. Furthermore, miR-30b-5p targeting KIF18A promoted PCa radiosensitivity by blocking autophagy. Taken together, circ-CCNB2 can bind to miR-30b-5p to affect KIF18A expression and regulate PCa radiosensitivity.

### 3.7. Materials and Methods Used in CircRNA Studies

In these studies, samples were obtained from patients. All mediums contained 10% fetal bovine serum (FBS; Gibco) and 1% penicillin/streptomycin (Invitrogen, Carlsbad, CA, USA), and all cells were incubated at 37 °C with a 5% CO_2_ incubator. Then, cell transfection was performed using Lipofectamine 3000 (Invitrogen).

After the isolation of total RNA from tissues and cells by TRIzol^TM^ Reagent (Invitrogen), cDNA was synthesized using cDNA Synthesis SuperMix. Quantitative analysis was carried out using SYBR Green on Real-Time PCR System in order to detect the circRNA expression. After transfection and treatment with X-ray irradiation was conducted for respective 0 min, 1 min, 2 min, and 4 min (0 Gy, 2 Gy, 4 Gy and 8 Gy), count; further, the colonies were counted under a microscope. Flow cytometry was performed to measure the cell cycle distribution and apoptosis of cells. Western blot was conducted for protein expression analysis. Construction of the luciferase plasmids was implemented by cloning the wild-type (WT) and mutant-type (MUT) sequences of circRNA into the vector (Promega, Madison, WI, USA), after which the relative luciferase activity was measured using the dual-luciferase reporter system (Promega).

## 4. Conclusions 

In the study of human cancer, the interactions between different RNAs are quite complex [153].

One circRNA can suppress the expression levels of different miRNAs and thus affect the expression and function of mRNAs. For example, in addition to regulating tumor resistance by sponging miR-329-3p [64], circPITX1 can also regulate ERBB4 expression by sponging miR-1304, thus promoting glioma progression [154]. In addition, circPITX1 promotes glioblastoma evolution by regulating MAP3K2 as a competitive endogenous RNA through the uptake of miR-379-5p [155].

A circRNA can act in different tumors. Knockdown of circ-0086720 enhances the radiosensitivity of NSCLC cells [69]. At the same time, circ-0086720 is highly expressed in radiation-resistant esophageal cancer cells [156], suggesting that circ_0086720 may play an important role in the radiation resistance of esophageal cancer. Circ-PVT1 has been reported as a proliferative factor in many cancers, such as gastric [157], colorectal [158], and esophageal [159] cancers. In gastric cancer (GC), circ-PVT1 downregulation impaired chemotherapy resistance to paclitaxel [160]. In lung adenocarcinoma, circ-PVT1 overexpression is involved in chemoresistance through the miR-145-5p/ABCC1 axis [161].

In conclusion, circRNAs can affect the proliferation, migration, and invasion of irradiated tumor cells and inhibit apoptosis through multiple pathways. It provides a new idea to attenuate the radioresistance of tumor cells.

## 5. Discussion and Prospects

Radiation therapy developed rapidly following Roentgen’s discovery of X-rays, Becquerel’s discovery of natural radioactivity, and Curie’s discovery of radium. Currently, radiotherapy is one of the main methods of treating cancer [17]. However, tumor radioresistance remains a crucial barrier to treatment outcomes. 

Biomarkers can be used to aid cancer diagnosis, determine prognosis, and monitor progression. In general, biomarkers should have good sensitivity, specificity, and stability [162]. Further, circRNAs are expressed explicitly in tissues while being structurally stable [6]. In addition, circRNAs are enriched in human body fluids such as saliva [163] and blood, making them easy to detect, and these characteristics make circRNAs ideal as biomarkers. Most of the circRNAs we mentioned are highly expressed in tumor cells and are also associated with radioresistance of tumors, which means that circRNAs can be used not only as diagnostic biomarkers, but also as biomarkers for determining prognosis.

CircRNAs hold promise as therapeutic targets through circRNA loss-of-function therapy or circRNA gain-of-function therapy. Antisense technologies can be used to inhibit or degrade oncogenic circRNAs selectively. For example, complementary single-stranded DNA antisense oligonucleotides can be designed and annealed to a unique sequence in the targeted circRNA. Targeted cleavage of its site by intracellular RNaseH enzymes or RNA interference (RNAi) methods to induce cleavage of circRNAs [164], techniques that target circRNAs for disruption has been used to treat diseases other than tumors [165]. However, the challenges of delivering circRNAs to cells are a major clinical implementation obstacle [166]. These seem to explain why most studies have targeted up-regulated circRNAs rather than down-regulated circRNAs.

In particular, the articles we have summarized, circRNAs generally act as miRNA sponges [13]. Although the role of circRNAs as miRNA sponges is well established, other potential functions of circRNAs in radiation therapy require further investigation [14]. For circRNAs, studying the process of a group of significantly differentiated circRNAs or individual circRNAs and examining their potential targets could reveal more about the role of circRNAs in cancer progression. Furthermore, most reports on the part of circRNAs in tumor resistance to radiotherapy have been limited to a small number of cancers. In recent years, ferroptosis has been proposed and could also be used as a new way of thinking about treatment. Although research on circRNAs modulating tumor radiotherapy resistance is still in its infancy, we believe circRNAs can provide fresh ideas for tumor radiotherapy.

## Figures and Tables

**Figure 1 biomolecules-12-01586-f001:**
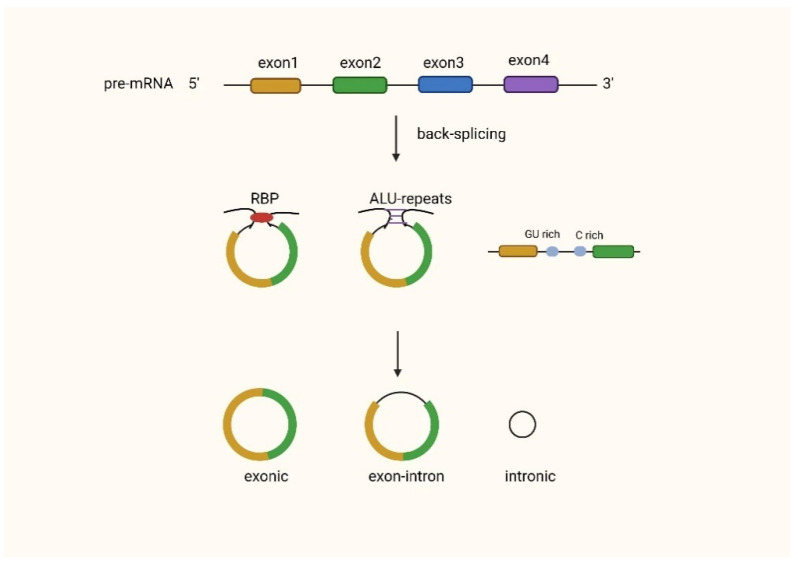
Biogenesis of circRNAs.

**Figure 2 biomolecules-12-01586-f002:**
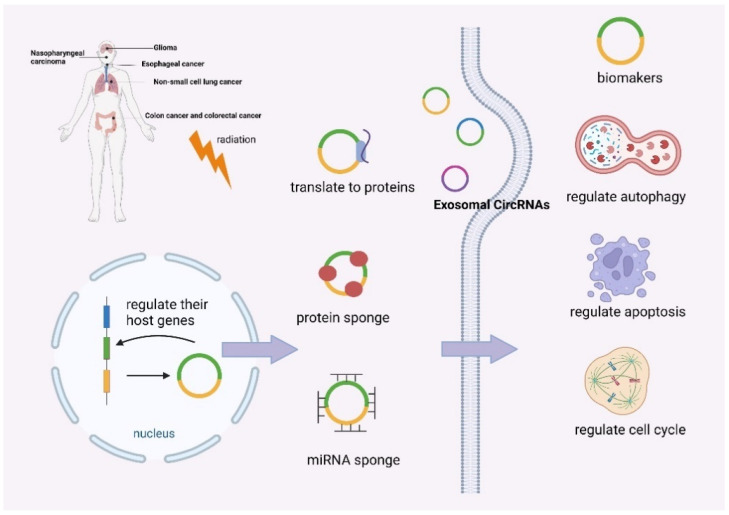
The mechanism of circRNA in tumor radioresistance.

**Table 1 biomolecules-12-01586-t001:** Alterations of circRNAs in radiotherapy resistance.

Cancer	Name of CircRNAs	Alteration	Target	Mechanism	References
glioma	circ-0008344	+	miR-433-3p/RNF2	enhance apoptosis and DNA damage	[62]
	circ-VCAN	+	miR-1183	enhance apoptosis	[63]
	circ-PITX1	+	miR-329-3p/NEK2	inhibit glycolysis	[64]
nasopharyngeal carcinoma	circ-000543	+	miR-9/PDGFRB	EMT	[65]
	circ-001387	+	Not mentioned	biomarker	[66]
	circ-0000285	+	Not mentioned	biomarker	[67]
non-small cell lung cancer	circ-0001287	−	miR-21/PTEN	regulate cell cycle progression and DNA damage	[68]
	circ-0086720	+	miR-375/SPIN1	enhance apoptosis	[69]
	circ-PVT1	+	miR-1208	EMT	[70]
	circ-ZNF208	+	miR-7-5p/SNCA	regulate apoptosis	[71]
colon cancer	circ-0055625	+	miR-338-3p/MSI1	regulate apoptosis	[72]
	circ-CCDC66	+	miR-338-3p	enhance apoptosis	[73]
colorectal cancer	circ-0007031	+	miR-760/DCP1A	regulate cell cycle	[74]
	circ-0067835	+	miR-296-5p/IGF1R	regulate cell cycle progression and DNA damage	[75]
	circ-MFN2	+	miR-574-3p/IGF1R	regulate cell cycle progression	[76]
	circ-ACAP2	+	miR-143-3p/FZD4	regulate apoptosis	[77]
	circ-CBL.11	+	miR-6778-5p	DNA damage	[78]
	circ-IFT80	+	miR-296-5p/MSI1	regulate cell cycle progression	[79]
esophageal cancer	circ-PRKCI	+	miR-186-5p/PARP9	regulate cell cycle progression	[80]
	circ-0000554	+	miR-485-5p/FERMT1	regulate apoptosis	[81]
	circ-VRK1	−	miR-624-3p/PTEN/PI3K/AKT	EMT	[82]
	circ-0014879	+	miR-519-3p/CDC25A	regulate cell cycle progression and DNA damage	[83]
prostate cancer	circ-0062020	+	miR-615-5p/TRIP13	enhance apoptosis	[84]
	circ-CCNB2	+	miR-30b-5p/KIF18A	repressing autophagy	[85]
	circ-ZNF609	+	miR-501-3p/HK2	inhibit glycolysis	[86]
cervical cancer	circ-0009035	+	miR-889-3p/HOXB7	regulate apoptosis	[87]
breast cancer	circ-ABCB10	+	miR-223-3p	inhibit glycolysis	[88]

“+” means the circRNA was upregulated and “−” means the circRNA was downregulated.

## Data Availability

Not applicable.

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
