# Peer review of "CircRNAs in Tumor Radioresistance"

_biomolecules, 2022, doi:10.3390/biom12111586_

Round 1

Reviewer 1 Report

The present study aims to summarize the circRNAs that are implicated in tumor radioresistance. The circRNAs have attracted researchers’ interest due to their key role in normal cell function and in the development of pathological states. Therefore, the topic is quite interesting. However, the present review has the following major issues that have to be addressed prior to its publication:

1.      The section of introduction should be extended and more information regarding the circRNA biology should be included. For instance, circRNAs derive from both pre-mRNA and pre-tRNA molecules.

2.      The present study contains only 1 figure. More figures should be included so that this review is more attractive to the readers.

3.      Is there additional literature regarding the biomarker utility of the circRNAs in this research field?

4.       The authors have divided the review based on the cancer type and mention circRNAs that have been implicated in this malignancy. However, they do not analyze these findings in depth and do not provide a solid conclusion.

5.      The authors do not provide the future perspectives of this research field. Additionally, the authors should include in the manuscript the limitations of the field.

6.      The authors should clarify in each research included in the review in what type of material the experiments were conducted.

7.      Are there circRNAs which have been reported in more than one type of malignancy?

8.      The authors could include an additional section in which they analyze the exosomal circRNAs more in depth.

9.      In the Table 1, the gene origin of each circRNAs should be included.

Author Response

We are very grateful for your comments, and we have taken most of them on board and highlighted in red what we have changed. As for the other elements you mentioned, such as the origin of each circRNA, we are sorry that we did not find the relevant information in the article.

Reviewer 2 Report

A sufficient number of papers are included and cited in this review.

The summary (abstract) seems to be too short to understand the importance of this review.

The mechanism of (most) upregulation of circRNAs in radiation-resistant tumors are unknown or unclear. Why almost all circRNAs are upregulated by radiation? This reviewer considers this regulation is very important. If some mechanisms are verified in a particular study, it should be clearly described and discussed. If not, it should be noted as such in a prospective section.

For the description of specific miRs, "miR-" and "miR" are used together in the text.

Likewise, ‘circ_’, ‘circ-‘and ‘circ’ are mixedly used. Please unify.

Citation of 73 is not appropriately described.

Do the descriptions of '+' in the table mean upregulation and '-' downregulation? Some explanation is better to added.

This is also true to Fig 1. A figure legend including some explanation should be provided.

The significance of the finding differs between the results of in vitro and in vivo studies.  Similarly, it is useful to know form the table whether these circRNAs were detected in human samples such as tumors or blood. Please include these information in the table.

Author Response

We appreciate your comments and have added the contents you mentioned and highlighted them in red.

Round 2

Reviewer 1 Report

In the revised manuscript the Authors answered some comments; however, the majority of them were not sufficiently addressed.  Specifically, the following comments have to be addressed prior to publication:

1.      The section of the introduction should be extended and more information regarding circRNA biology should be included. For instance, circRNAs derive from both pre-mRNA and pre-tRNA molecules.

2.      The present study contains only 1 figure. More figures should be included to make this review more attractive to the readers.

3.      The authors have divided the review based on the cancer type and mention circRNAs that have been implicated in this malignancy. However, they do not analyze these findings in depth and do not provide a solid conclusion.

4.      The authors do not provide the future perspectives of this research field. Additionally, the authors should include in the manuscript the limitations of the field.

5.      The authors should clarify in each research included in the review in what type of material the experiments were conducted.

6.      Are there circRNAs that have been reported in more than one type of malignancy?

Author Response

1.We added that "circRNAs derive from both pre-mRNA" in both"Introduction" and  "Biogenesis of circRNAs". However, we did not find any article containing circRNAs derive from pre-tRNA.

2.We added figures, one of which is at the location of the Biogenesis of circRNAs and the other is below the table.

3.Now, we have added a natural paragraph "conclusion" to clarify our conclusions.

4.According to the comments made by the reviewer, we conclude the article by clarifying the limits in this area as well as our outlook.

5.We summarize the methods mentioned in the study with circRNA and add this section to the end of "the Research progress of circRNAs related to tumour radioresistance".

6.There are some circRNAs that have been reported in more than one type of malignancy,they help with the diagnosis and treatment of tumors through different pathways. We also added this part to the end of the article.